# Peer review of "Graphite/β-PbO2 Composite Inert Anode Synthesis Using Electrochemical Methods"

_2305-7084, doi:10.3390/chemengineering7020020_

Round 1
Reviewer 1 Report (New Reviewer)
The reviewed article Graphite/β-PBO2COMPOSITE INERT ANODE SYnthesis Using Electrocal Methods Recperezn will be a high scientific level. Each of the presented parts of the article was developed and presented in a very clear way. The whole article is consistent, closely related and presents the results of high scientific value. To sum up, each part of the publication has been described in detail, the charts presented, the diagrams are very clear to recipients. The conclusions are consistent and closely related to the topic of research. As a reviewer of this work, I will not pay any comments. I believe that the authors have exhausted all topics in the reviewed work.
Author Response
Response: Thank you so much
Reviewer 2 Report (Previous Reviewer 2)
The title of this manuscript is “Graphite/β-PbO2 composite inert anode synthesis using electrochemical methods”. The article is innovative and addresses scientific issues. Authors fails to present the data obtained, and no explanations for adsorption was provided. Therefore, authors should do the following major issues before publishing in Chem Engineering.
11. This manuscript is poorly presented and contains several grammatical errors.
22. The introduction section contains so many paragraphs. It should be avoided.
33. Authors should check the formula equations and chemical reactions.
44. How did the authors claim about the inertness of this electrode materials? NO explanation has been carried out.
55. Authors should explain the XRD and XPS analysis to understand the formation of Graphite/β-PbO2 composite.
66. Why did authors choose β-PbO2?
77. The following references using the examples of GO/transition metal composite synthesis given below can be cited in appropriate places, such as 10.1016/j.cej.2022.138471; 10.3390/catal10050546
88. Mention the mass of materials deposited at cathode.
99. The reference format is not consistent and authors should double check.
Author Response
Response to Reviewer 2 Comments
The title of this manuscript is “Graphite/β-PbO2 composite inert anode synthesis using electrochemical methods”. The article is innovative and addresses scientific issues. Authors fails to present the data obtained, and no explanations for adsorption was provided. Therefore, authors should do the following major issues before publishing in Chem Engineering.
- This manuscript is poorly presented and contains several grammatical errors.
Response 11:
This article “Graphite/β-PbO2 composite inert anode synthesis using electrochemical methods” is an improvement from the previously submitted article ChemEngineering-1744236 entitled “High Current Density to Increase Amount of β -PbO2 Using Electrochemical Methods”. Most of the contents of the article are the same as the previous article, the article has been improved grammatically and writing style (Id: 46252) as evidenced by:
- The introduction section contains so many paragraphs. It should be avoided.
Response 22: The introduction has been improved from 10 paragraphs (814 letters) to 5 paragraphs (556 letters).
- Authors should check the formula equations and chemical reactions.
Response 33: Equation 1 corrosion rate has been fixed to:
Reaction:
- How did the authors claim about the inertness of this electrode materials? NO explanation has been carried out.
Response 44:
An inert electrode is an electrode that only functions as a source or reservoir of electrons without acting chemically to react as reactants or products or act as an electrocatalyst. The precious metals, Pt and C are usually used as inert electrodes.
Based on the reference "Inert Anode Technologies Report", the constituent materials of the inert anode consist of cermets (a combination of two materials, namely metal and ceramic), metal oxide (ceramic), metal. In this study, we used a graphite substrate coated with PbO2 (metal oxide/ceramic). Based on the composition used, the material formed is inert.
- Authors should explain the XRD and XPS analysis to understand the formation of Graphite/β-PbO composite.
Response 55: We did this research in our department, but for the Tafel polarization test and SEM-EDS, it is not available in our department. We send samples to places where Tafel polarization testing is available at Technology Assessment and Application Agency and SEM-EDS at Chemical Engineering Surabaya Institute of Technology. For that analysis we have sent all our samples, so to do XRD testing, we can't do it because we must do our research from beginning.
I am so sorry, in this article we have not been able to fulfill the results of the XRD analysis, due to the limited analytical equipment available in our department.
- Why did authors choose β-PbO2?
Response 66: The choice of lead dioxide as a metal oxide electrode because it is inexpensive, stable and has a strong structure for anodic oxidation at a very positive potential.
From a microcosmic point of view, the lead dioxide layer consists of different phase structures, namely - α-PbO2 and β-PbO2 significantly resulting in different catalytic activities. Orthorhombic α-PbO2 is a compact structure with close contact between particles and is chemically more stable. The tetragonal - β-PbO2 is an open porous structure that provides a larger active surface area. β-PbO2 is much more electrocatalytically active, surface β-PbO2 has higher performance.
- The following references using the examples of GO/transition metal composite synthesis given below can be cited in appropriate places, such as 10.1016/j.cej.2022.138471; 10.3390/catal10050546
Response 77: Thank you for suggesting references related to composite synthesis, very helpful.
- Mention the mass of materials deposited at cathode.
Response 88: It was discussed in 3.2. and 4.2 Effect of Voltage and Sulfuric Acid Concentration on Cathode Mass Increase
- The reference format is not consistent, and authors should double check.
Response 99: Fixed the format of references and writing compounds.
Response to Reviewer 2 Comments
The title of this manuscript is “Graphite/β-PbO2 composite inert anode synthesis using electrochemical methods”. The article is innovative and addresses scientific issues. Authors fails to present the data obtained, and no explanations for adsorption was provided. Therefore, authors should do the following major issues before publishing in Chem Engineering.
- This manuscript is poorly presented and contains several grammatical errors.
Response 11:
This article “Graphite/β-PbO2 composite inert anode synthesis using electrochemical methods” is an improvement from the previously submitted article ChemEngineering-1744236 entitled “High Current Density to Increase Amount of β -PbO2 Using Electrochemical Methods”. Most of the contents of the article are the same as the previous article, the article has been improved grammatically and writing style (Id: 46252) as evidenced by:
- The introduction section contains so many paragraphs. It should be avoided.
Response 22: The introduction has been improved from 10 paragraphs (814 letters) to 5 paragraphs (556 letters).
- Authors should check the formula equations and chemical reactions.
Response 33: Equation 1 corrosion rate has been fixed to:
Reaction:
- How did the authors claim about the inertness of this electrode materials? NO explanation has been carried out.
Response 44:
An inert electrode is an electrode that only functions as a source or reservoir of electrons without acting chemically to react as reactants or products or act as an electrocatalyst. The precious metals, Pt and C are usually used as inert electrodes.
Based on the reference "Inert Anode Technologies Report", the constituent materials of the inert anode consist of cermets (a combination of two materials, namely metal and ceramic), metal oxide (ceramic), metal. In this study, we used a graphite substrate coated with PbO2 (metal oxide/ceramic). Based on the composition used, the material formed is inert.
- Authors should explain the XRD and XPS analysis to understand the formation of Graphite/β-PbO composite.
Response 55: We did this research in our department, but for the Tafel polarization test and SEM-EDS, it is not available in our department. We send samples to places where Tafel polarization testing is available at Technology Assessment and Application Agency and SEM-EDS at Chemical Engineering Surabaya Institute of Technology. For that analysis we have sent all our samples, so to do XRD testing, we can't do it because we must do our research from beginning.
I am so sorry, in this article we have not been able to fulfill the results of the XRD analysis, due to the limited analytical equipment available in our department.
- Why did authors choose β-PbO2?
Response 66: The choice of lead dioxide as a metal oxide electrode because it is inexpensive, stable and has a strong structure for anodic oxidation at a very positive potential.
From a microcosmic point of view, the lead dioxide layer consists of different phase structures, namely - α-PbO2 and β-PbO2 significantly resulting in different catalytic activities. Orthorhombic α-PbO2 is a compact structure with close contact between particles and is chemically more stable. The tetragonal - β-PbO2 is an open porous structure that provides a larger active surface area. β-PbO2 is much more electrocatalytically active, surface β-PbO2 has higher performance.
- The following references using the examples of GO/transition metal composite synthesis given below can be cited in appropriate places, such as 10.1016/j.cej.2022.138471; 10.3390/catal10050546
Response 77: Thank you for suggesting references related to composite synthesis, very helpful.
- Mention the mass of materials deposited at cathode.
Response 88: It was discussed in 3.2. and 4.2 Effect of Voltage and Sulfuric Acid Concentration on Cathode Mass Increase
- The reference format is not consistent, and authors should double check.
Response 99: Fixed the format of references and writing compounds.

Reviewer 3 Report (New Reviewer)
The article raphite/β-PbO2 Composite Inert Anode Synthesis Using Electrochemical Methods is devoted to the study of the properties of anode materials for batteries. The presented results have a certain level of novelty, and the article corresponds to the subject of the declared journal. According to the reviewer, the article can be accepted for publication after the authors answer a number of questions, including both the correction of the presented data and the explanation of the observed effects.
1. In the abstract, the authors should indicate the appropriateness of choosing the type of anode material.
2. The authors should give a more detailed description of the presence of inclusions on the surface of the samples, in the form of cubic particles, as well as their number depending on the number of test cycles. Whether their presence is associated with the formation of oxides or degradation of the surface.
3. The results of energy dispersive analysis look quite convincing, but the authors should focus on the use of other methods of analysis, including the X-ray diffraction method and the morphological features of the samples.
4. The authors should describe in more detail the polarization curves, as well as the change in their nature depending on the composition of the anode.
Author Response
Response to Reviewer 3 Comments
The article raphite/β-PbO2 Composite Inert Anode Synthesis Using Electrochemical
Methods is devoted to the study of the properties of anode materials for batteries. The
presented results have a certain level of novelty, and the article corresponds to the subject
of the declared journal. According to the reviewer, the article can be accepted for
publication after the authors answer a number of questions, including both the correction
of the presented data and the explanation of the observed effects.
1. In the abstract, the authors should indicate the appropriateness of choosing the type of
anode material.
Response 1:
Abstract: The anode material is one of the determining factors for the success of the
electrowinning process. This study aims to coat the graphite substrate with β-PbO2 to
produce an inert graphite/ β-PbO2 composite material with low cost and good quality.
The graphite/β-PbO2 synthesis is expected to be used as an anode for inert composites for
electrowinning processes. The β-PbO2 deposition layer was prepared on the surface of
the graphite substrate by an electrodeposition process using a sulfuric acid electrolyte.
The effect of electrolyte concentration and voltage on graphite/ β-PbO2 synthesis was
investigated using a potentiodynamic polarization test through Tafel analysis.
Experimental data have shown that there is an increase in the current density value at the
beginning of the process and then a decrease due to passivation, this is due to anodic
polarization in the graphite/β-PbO2 synthesis process. Suitable conditions can effectively
increase the rate of formation of β-PbO2. The results of scanning electron microscopy with
energy dispersive X-ray spectroscopic analysis of the formed crystal structure showed
that the β-PbO2 deposition layer obtained had a well-formed tetragonal structure at a
voltage of 9 V.
2. The authors should give a more detailed description of the presence of inclusions on
the surface of the samples, in the form of cubic particles, as well as their number
depending on the number of test cycles. Whether their presence is associated with the
formation of oxides or degradation of the surface.
Response 2: To explain in more detail the presence of inclusions, particle shape and
number of test cycles requires further analysis while all the samples we produce have
been sent for potentiodynamic testing and scanning electron microscopy with energy
dispersive X-ray spectroscopy analysis. I'm sorry, I can't explain in more detail because
of the limited testing equipment facilities at our institution.
3. The results of energy dispersive analysis look quite convincing, but the authors should
focus on the use of other methods of analysis, including the X-ray diffraction method and
the morphological features of the samples.
Response 3: We did this research in our department, but for the Tafel polarization test
and SEM-EDS, it is not available in our department. We send samples to places where
Tafel polarization testing is available at Technology Assessment and Application Agency
and SEM EDS at Chemical Engineering Surabaya Institute of Technology. For that
analysis we have sent all our samples, so to do XRD testing, we can't do it because we
must do our research from beginning.
I am so sorry, in this article we have not been able to fulfill the results of the XRD analysis,
due to the limited analytical equipment available in our department.
4. The authors should describe in more detail the polarization curves, as well as the
change in their nature depending on the composition of the anode.
Response 4:
The polarization curve shows the relationship of concentration and voltage in the
formation of and PbO2 coating on graphite. At the beginning of the electrodeposition
process there was an increase in the amount of PbO2 at the cathode (graphite) then it was
constant due to the formation of PbO2 material. The process of formation of PbO2 can be
explained by the following reaction:The Pb anode was active so that Pb would dissolve
in the electrolyte solution. The sulfuric acid used as an electrolyte solution acted as a
strong oxidizing agent causing Pb to be oxidized to unstable Pb. Pb, originating from the
anode surface reacted with negative ions SO4
2-
to form PbSO4. The oxidation reaction
occurred with Pb electrons at the anode surface and the sulfate ions in the electrolyte,
producing PbSO4 (1) and the oxygen evolution reaction, namely the decomposition of
water and liberation of oxygen (2).
?? + ??4
2− + 2? → ????4 (1)
2?2? → 4?
+ + ?2 + 4? (2)
At the cathode, there was a reduction in Pb ions from the solution, which moved towards
the cathode while receiving two electrons, and settled into Pb atoms on the
surface/coating the cathode. Then the Pb ions settled, and H2 ions entered the solution so
that the charge was maintained in the solution. This process continued continuously,
which could be observed in the form of gas bubbles and the changing color of the
solution.
????4 + 2?2? − 2? → ???2 + ?2??4 + 2?
+ (3)
The reaction between PbSO4 and H2O produced PbO2. The formation of this oxide layer
made the layer passive with corrosion resistance.

This manuscript is a resubmission of an earlier submission. The following is a list of the peer review reports and author responses from that submission.
Round 1
Reviewer 1 Report
The β-PbO2 was electrodeposited by potentiodynamic polarization, the applied potential and electrolyte concentration was investigated to achieve higher deposition current. The SEM-EDS was used to characterize the prepared β-PbO2. The author need to address following questions to improve the overall quality.
- 17-18 abstract; the “optimum value”: the author needs to specify what kind of optimum value. It is unclear for audience about this sentence.
- 23-24 abstract; the author claimed that PbO2 can be used as catalysts, but there are no such investigation/experiments in the entire article. That claim needs to be removed if no evidence to support.
- 53-54, 60-62; the author needs to make sure use same font for entire article. Please check entire article.
- The authors need to specify the reason why they choose to investigate the electrodeposition of PbO2, what is the current progress of such topic? How industry or academic synthesis PbO2? What is the advantage of electrochemical method compared with other synthesis routine? The authors need to reduce the discussion/introduction of limonene, because this is not the topic of this article. Need to rewrite the introduction part.
- 95-96: need to specify the potential (vs. NHE) of refence electrode
- 109-110: what lower case k stand for?
- Section 2.3.2, please provide more details about the SEM tool. For example, which electron voltage was used?
- Please label Figure 2 properly, it’s unclear what these lines stand for?
- Please check the caption of the figures, need to rework on it.
- 150-151; why the distribution not even? It is unclear to me why the carbon is not distribute everywhere because the FGC sheet was used.
- Reaction 3, it should form H2 instead of H+ to maintain the balance of the electrons
Author Response
REVIEWER 1
- 17-18 abstract; the “optimum value”: the author needs to specify what kind of optimum value. It is unclear for audience about this sentence.
The optimum value of current density was obtained at the concentration variations of 4 M and 9 V of 28.14 μA/cm2 and the quantity of material deposited was 33.03 mpy.
Revised
In this study, the highest current density value was obtained at a concentration variations of 4M and 9 Volt of 28.14 A/cm2 and the quantity of material deposited was 33.03 mpy.
- 23-24 abstract; the author claimed that PbO2 can be used as catalysts, but there are no such investigation/experiments in the entire article. That claim needs to be removed if no evidence to support.
Revised
23-24 abstract : Already deleted
- 53-54, 60-62; the author needs to make sure use same font for entire article. Please check entire article.
Revised
- The authors need to specify the reason why they choose to investigate the electrodeposition of PbO2, what is the current progress of such topic? How industry or academic synthesis PbO2? What is the advantage of electrochemical method compared with other synthesis routine? The authors need to reduce the discussion/introduction of limonene, because this is not the topic of this article. Need to rewrite the introduction part.
Revised
- 95-96: need to specify the potential (vs. NHE) of refence electrode
Revised
The actual potential of half-cell is +0.197 V vs SHE.
- 109-110: what lower case K stand for?
Revised
- Section 2.3.2, please provide more details about the SEM tool. For example, which electron voltage was used?
Revised
- Please label Figure 2 properly, it’s unclear what these lines stand for?
Revised
- Please check the caption of the figures, need to rework on it.
Revised

Reviewer 2 Report
The manuscript titled " High current density to increase amount of β-PbO2 using electrochemical methods" by Selpiana et al. reported that, electrodeposition of the β-PbO2 on flexible graphite sheets in the presence of different concentrations of sulfuric acid electrolyte. Furthermore, the authors studied the morphology analysis of the deposited β-PbO2 and reported the corrosion rate. The authors do not provide details on the results and discussion about the formation of the β-PbO2 and do not provide confirmation results of β-PbO2. Overall manuscript presentation is very poor. Therefore, the present manuscript should be need major revision for publication.
General comments:
- Authors should modify the title of the paper and abbreviations are missing in abstract. For example, FGS and SEM-EDS. Furthermore, authors should check the manuscript throughout.
- The introduction section's first three paragraphs describe the importance, usage, and synthesis of limonene. After that, the author discussed the electrodeposition of PbO2. There is no connectivity between limonene and the electrodeposition method of the present work. Therefore, authors should rewrite the introduction section with the present work's novelty and objective. Authors can refer following references such as 10.1002/bkcs.10379 and 10.1021/la0208068.
- There are many similar works available in literature. What is the novelty of the present work?
- The authors should check the format and captions of all the figures because the format and captions are uniform and not clear in figures.
- The scale bar on SEM images is not visible.
- How did you confirm the formation of the β-PbO2 after electrodeposition? should provide strong evidence.
Author Response
REVIEWER 2
1. Authors should modify the title of the paper and abbreviations are missing in abstract. For example,
FGS and SEM-EDS. Furthermore, authors should check the manuscript throughout.
Revised
2. The introduction section's first three paragraphs describe the importance, usage, and synthesis of
limonene. After that, the author discussed the electrodeposition of PbO . There is no connectivity
between limonene and the electrodeposition method of the present work. Therefore, authors
should rewrite the introduction section with the present work's novelty and objective. Authors can
refer following references such as 10.1002/bkcs.10379 and 10.1021/la0208068.
Revised
3. There are many similar works available in literature. What is the novelty of the present work?
Revised
4. The authors should check the format and captions of all the figures because the format and captions
are uniform and not clear in figures.
Revised
5. The scale bar on SEM images is not visible.
Revised
6. How did you confirm the formation of the β-PbO after electrodeposition? should provide strong
evidence.
Revised

Reviewer 3 Report
The paper is an electrochemical investigation about a possible route to improve the amount of PbO2 deposited by electrodeposition process. The topic could be interesting, nevertheless the manuscript is quite difficult to read, and should be improved in its form.
Line 28-29: in the first sentence is really hard to find the accordance between the subject and the verb. Check the form.
General the introduction is really difficult to read and follow, please improve the readability of this section. E.g. what does it mean "Electrochemical is a constantly evolving method for the synthesis of metal oxides." ? The english must be reviewed.
EXPERIMENTAL
Line 79: Please add the grain of sandpaper
Line 82:"rinsed with ethanol (Merck) for 3". For 3 what? please add the unit and then the point to conclude the sentence
Line 108: which is the molarity of Ag/AgCl electrode?
In general the paper should be revised in the form to valorize better the content
Author Response
The manuscript has been corrected according to the results of English editing (ID: English-46252)

Round 2
Reviewer 1 Report
1. The manuscript still miss the review of the previous electrochemical synthesis of PbO2. So the motivation of this study is still unclear, for example: why we test under different potential? and how this study will impact the overall scientific world or electrochemical synthesis?
2. Not all revised parts are highlighted correctly and not indicate in the response.
Author Response
This study was conducted with variations in electrolyte voltage and concentration to determine its effect
on increasing current density, if the current density increases, the material formed at the cathode also
increases.
Cathode mass gain is presented in 3.2 and its discussion on 4.2

Reviewer 2 Report
The manuscript is improved to satisfaction. However, many mistakes were found in the Text and the format of Reference of the manuscript. Please check again.
Author Response

(The authors gave the same response as above.)

Round 3
Reviewer 1 Report
My question still not addressed properly. It's still not clear about the motivation(s) of this study, they didn't review previous studies very well. It's more like a scientific/lab report at graduate student level.